# A Crack Segmentation Model Combining Morphological Network and Multiple Loss Mechanism

**DOI:** 10.3390/s23031127

**Published:** 2023-01-18

**Authors:** Fan Zhao, Yu Chao, Linyun Li

**Affiliations:** Department of Information Science, Xi’an University of Technology, Xi’an 710048, China

**Keywords:** crack segmentation, U-Net network, morphological network

## Abstract

With the wide application of computer vision technology and deep-learning theory in engineering, the image-based detection of cracks in structures such as pipelines, pavements and dams has received more and more attention. Aiming at the problems of high cost, low efficiency and poor detection accuracy in traditional crack detection methods, this paper proposes a crack segmentation network by combining a morphological network and a multiple-loss mechanism. First, for improving the identification of cracks with different resolutions, the U-Net network is used to extract multi-scale features from the crack image. Second, for eliminating the effect of polarized light on the cracks under different illuminations, the extracted crack features are further morphologically processed by a white-top hat transform and a black-bottom hat transform. Finally, a multi-loss mechanism is designed to solve the problem of the inaccurate segmentation of cracks on a single scale. Extensive experiments are carried out on five open crack datasets: Crack500, CrackTree200, CFD, AEL and GAPs384. The experimental results showed that the average ODS, OIS, AIU, sODS and sOIS are 75.7%, 73.9%, 36.4%, 52.4% and 52.2%, respectively. Compared with state-of-the-art methods, the proposed method achieves better crack segmentation performance. Ablation experiments also verified the effectiveness of each module in the algorithm.

## 1. Introduction

Inevitably, there are some injuries and defects in concrete structures that arise in the process of building and using because of reasons having to do with design, construction, loads and materials and so on. These defects, which initially present as cracks, should seriously affect the safety and durability of a structure, and they need to be discovered as early as possible. Crack segmentation based on machine vision is a process of automatically extracting cracks from images captured by cameras. Automatic crack segmentation not only has important theoretical research significance, but also has an important early warning application in practical engineering.

Crack segmentation methods based on machine vision are mainly divided into traditional algorithms [1,2,3,4,5,6,7,8,9] and deep-learning-based methods [10,11,12,13,14,15,16,17,18,19,20,21,22,23,24,25,26,27,28,29,30,31,32,33].

Compared with the depth-learning method, the traditional crack segmentation method does not require training and is simpler and more direct [1,2,3,4,5,6,7,8,9]. This kind of algorithm is easy to implement and fast in operation. Although the traditional method is simple and fast, it is easily affected by noise. Improper threshold selection will significantly reduce the segmentation accuracy.

Depth-learning-based methods can automatically extract the shallow and deep characteristics of cracks, so they can greatly improve segmentation and detection accuracy with regard to cracks. The backbone networks commonly used for crack segmentation include the multi-scale pyramid structure network [10,11,12,13,14,15,16,17] and the U-Net network [18,19,20,21,22]. Methods based on multi-scale features can extract more abundant crack image features, but the existing methods make it difficult to detect cracks similar to the background. Because of its good performance in medical image segmentation, the U-Net network and its expansion are widely used in crack detection [18,19,20,21,22]. From the experimental results of many studies, it can be seen that the methods based on deep learning have a significant improvement in the effects of crack segmentation. However, the segmentation accuracy of cracks under polarized light is generally poor.

In order to improve crack segmentation performance under polarized light, this paper proposes a crack segmentation method that combines a morphological network and a multi-loss function. In summary, the main contributions of this paper are listed as follows.

(1) In view of inaccurate crack detection caused by the lack of an effective processing mechanism for the influence of the existing depth network model on polarized light, this paper proposes a morphological-processing network module, which processes the crack feature with a white-top hat and a black-bottom hat transformation to correct the influence of uneven lighting, and the bright cracks on the dark background and the dark cracks under the bright background are effectively extracted by the proposed method.

(2) In view of the incomplete detection on different scales of the crack image, a multi-scale loss fusion mechanism is adopted to overcome the problem of too thin or too thick cracks extracted by single-scale detection. 

(3) The experimental results show that the proposed method can accurately detect all kinds of cracks in the five public crack datasets.

The paper is organized as follows. Related works are presented in Section 2. The proposed crack segmentation method is proposed in Section 3. Experimental results and analysis are presented in Section 4. The conclusions are drawn in Section 5. 

## 2. Related Works

Compared with deep-learning methods, traditional crack segmentation methods do not require training, and the execution process is simpler and more straightforward [1,2,3,4,5,6,7,8]. In traditional methods, manual features are usually used to extract cracks. Because cracks exhibit certain edge characteristics, commonly used edge detection operators are performed for the extraction of cracks [1]. Based on Euclidean graphs as crack pattern descriptors, salient skeleton features are extracted to locate the cracks in an image [5]. A Gabor filter is used to detect the cracks in any direction [8]. Morphological filters and dynamic thresholding are applied to extract cracks of different thicknesses [9]. Using the pixel brightness difference between cracks and surrounding areas, methods such as random structure forest [4] and the threshold-processing method [6,7] can also effectively extract cracks. Although the traditional method is simple in implementation and fast in operation, it is easily affected by noise. Improper selection of the threshold will significantly reduce the segmentation accuracy.

Because the shallow and deep features of an image can be extracted automatically, the method based on deep learning can greatly improve the accuracy of crack segmentation and detection. Due to the ability to extract the information at different resolutions, multi-scale features are widely used in the segmentation and detection of cracks of different sizes [10,11,12,13,14,15,16,17]. Chun et al. [10] proposed a pavement crack detection method based on multi-scale attention and hesitant fuzzy set (HFS) theory. Saining et al. [11] first performed edge detection with a multiple loss structure, and they then fused multi-scale features to improve the detection effect of cracks. Yang et al. [12] proposed a pyramid structure and a hierarchical boosting network to detect pavement cracks. Wang et al. [13] proposed a bridge crack detection model, which combined the Inception-Resnet-v2 module, multi-scale feature fusion and a GKA clustering mechanism to improve the real-time detection performance. However, it is difficult for this algorithm to segment the small cracks accurately. Sun et al. [14] used an adaptive bilateral filtering algorithm to reduce the influence of noise before using the FPHBN network to segment the cracks. Li et al. [15] fused feature maps of different scales and used a class-balanced cross-entropy loss function to improve the accuracy, speed and robustness of crack segmentation, but the edge of the segmented crack was blurred. The attention mechanism is imposed on the fused multi-scale feature to enhance the distinction of crack feature representation [16]. A feature aggregation network with the spatial-channel squeeze and excitation attention mechanism module was proposed in paper [17] to accurately segment cracks. Although these methods based on multi-scale features can extract more abundant fracture image features, they were unable to distinguish cracks and backgrounds under biased light, which would lead to a decline in segmentation and detection performance.

Since the U-Net network can not only extract multi-scale features of objects with different sizes, the cross-layer connection in the encoder–decoder structure can also avoid information loss in the deep network. Therefore, the U-Net network has been widely used in image target segmentation, since it was proposed for medical image segmentation. Due to its excellent segmentation performance, U-Net network and its extension have also been widely used in crack segmentation [18,19,20,21,22] and rust detection in metallic constructions [23]. Hong et al. [20] introduced an attention module to the U-Net network and fused the features of skip connections to segment cracks on UAV aerial photography pavement. Jacob et al. [21] proposed an improved U-Net algorithm and introduced a data augmentation strategy to improve the accuracy of crack segmentation. A U-Net network with alternately updated clique was designed to separate cracks from the background [22].

In addition, scholars have also conducted research on crack segmentation and detection based on a combination of detection and segmentation [24,25], transformers [26], super-resolution reconstructions [27], an attention mechanism [28,29], transfer-learning technology [30], a fully convolutional network [31,32,33], deep learning and heuristic image post-processing [34]. As a result, relatively effective results have been obtained in the segmentation of various cracks.

Compared with traditional methods, the methods based on depth learning have achieved better crack segmentation performance, but due to the lack of an effective mechanism to remove polarized light, it is difficult to accurately detect cracks with an uneven brightness distribution.

Aiming at the problem of inaccurate crack detection caused by uneven illumination in images, this paper proposes a morphological network module, which performs white-top hat and black-bottom hat processing on the crack feature map to correct unevenness. Aiming at the incomplete description of cracks by the shallow and deep features of the network, a multi-scale loss fusion mechanism is used to overcome the problem of too thin or too coarse cracks extracted by single-scale detection.

## 3. The Proposed Crack Segmentation Method

A flow chart of the proposed crack segmentation method is shown in Figure 1, which consists of four main modules: feature extraction based on a backbone network, feature enhancement based on morphological processing, multi-scale feature fusion and multi-objective loss function calculation.

First, the crack image is inputted into the U-Net network to extract features of crack images. The U-Net network uses an encoder–decoder structure to extract features of different scales and fuses features of the same scale of encoder and decoder to make the crack feature more prominent. Second, the extracted features are inputted into the morphological-processing network, and the white-top hat and black-bottom hat transformation are performed, respectively, to correct the influence of uneven illumination on the captured crack images. Then, the output features of the U-Net network and the morphological-processing network are fused and inputted into the side network to obtain a crack prediction map at each scale. Finally, the multi-scale prediction results are fused to obtain the final crack segmentation result, and the loss function of each scale and the final prediction map loss are fused into the final loss function.

### 3.1. Multi-Scale Feature Extraction Based on U-Net Network

This paper selects the U-Net network as the feature extraction module. As shown in Figure 2, the U-Net network consists of the left encoding part, the right decoding part and the next two convolution and activation layers. The encoding part consists of four repeating structures, each of which consists of two 3 × 3 convolutional layers, nonlinear ReLU layers and a 2 × 2 max pooling layer, which correspond to the blue and red arrows in Figure 2, respectively. The decoding part is similar to the encoding part and also consists of four repeating structures. Deconvolution up-conv is used before each repeating structure to halve the number of channels and double the size of the feature map, corresponding to the green arrow in the figure. The deconvolution result is concatenated with the feature map of the same scale from the corresponding encoding part, which corresponds to the white/blue blocks of each repeating structure. The concatenated feature map is subjected to two 3 × 3 convolutions, which correspond to the blue arrows in the decoding part. At the last layer of the network, the feature map with 64 channels is converted into the prediction result of whether it is a crack through 1 × 1 convolution, corresponding to the cyan arrow. As shown in Table 1, the parameters of the U-Net network structure are listed.

In this paper, a pavement crack image with a size and dimension of W×H×3 is used as the input image of the U-Net network, and the three features of the decoder part with sizes of W×H×1, $\frac{W}{2}\times \frac{H}{2}\times 128$ and $\frac{W}{4}\times \frac{H}{4}\times 256$ are extracted and sent to the morphological network for depolarization processing.

### 3.2. Morphological Network

When the camera takes a picture of the crack, different parts of the crack perhaps appear as different colors and brightness values due to the change of light and shadow and different shooting angles, which is called a polarized light phenomenon. This phenomenon will lead to the same crack being treated as different objects, thus affecting the performance of target segmentation. In order to overcome the influence of polarized light on object segmentation performance, a morphological network is designed to enhance the output features of the U-Net network to highlight the bright cracks on a dark background and the dark cracks on a bright background.

The morphological network consists of the white-top hat transformation $T_{hat}\left(I\right)$ and the black-bottom hat transformation $B_{hat}\left(I\right)$:(1)$$T_{hat}\left(I\right)=I-\left(I⊖W_{e}\right)$$
(2)$$B_{hat}\left(I\right)=\left(I⊕\;W_{d}\right)-I$$
where $I\in R^{W\times H\times C}$ represents the output feature of the U-Net network, $\left(I⊖W_{e}\right)$ represents the morphological erosion operation on I and $\left(I⊕W_{d}\right)$ represents the morphological expansion operation on I. Here, $W_{d}\in R^{M\times N\times K}$ and $W_{e}\in R^{M\times N\times K}$ are dilation and erosion filters, and W×H and C represent the resolution and number of channels of the extracted features, respectively. M×N and *K* are the size and the number of the filters.

For ∀k∈[1,K], x∈[1, W], y∈[1, H], the dilation (⊕) and erosion (⊖) operations on the feature map I are as follows:(3)$$(I⊕W_{d})(x,y)=\underset{i\in S_{1}\;,j\in S_{2}}{\max}\left(I\left(x+i,y+j,k\right)+W_{d}\left(i,j,k\right)\;\right)$$
(4)$$(I⊖W_{e})(x,y)=\underset{i\in S_{1}\;,j\in S_{2}}{\min}\left(I\left(x+i,y+j,k\right)-W_{e}\left(i,j,k\right)\;\right)$$

The value ranges of $S_{1}$ and $S_{2}$ are as follows:(5)$$S_{1}=\left[-\frac{M-1}{2},\frac{M-1}{2}\right]$$
(6)$$S_{2}=\left[-\frac{N-1}{2},\frac{N-1}{2}\right]$$

Examples of dilation and erosion operations are shown in Figure 3 and Figure 4, respectively. The erosion operation is to make the dark area in the image larger, and the dilation operation is to make the bright area in the image larger. The functions of dilation and erosion are to eliminate noise, segment independent cracks, connect adjacent elements in the image and find the maximum or minimum area in the image. Therefore, subtracting morphologically manipulated images from the original image (and vice versa) can highlight areas that are brighter or darker than that in the original image, thus correcting the effects of uneven illumination. The structure diagram of the morphological network is shown in Figure 5, which consists of parallel erosion and dilation processing layers, a subtracting operation and a concatenation operation. The K feature maps of the expansion layer are obtained by convoluting K expansion filters $W_{d}$ with the output features of U-Net. In the same way, the K feature maps of the erosion layer are obtained by convolving the K erosion filters $W_{e}$ with the output features of the U-Net. After expansion and corrosion processing, the white-top hat feature and black-bottom hat feature are obtained through difference operation. The difference feature map and morphological processing feature map are concatenated in series, weighted by linear combination and convolved by a 1 × 1 filter to obtain the output feature map of the morphological network.

In order to verify the effectiveness of the morphological network module, feature maps before and after the morphological network processing are extracted, respectively, which are shown in Figure 6. As shown in Figure 6, there is a large difference between the extracted crack shape and the ground truth data without morphological processing, and there are many false detection areas. Compared to the feature map before morphological processing, the feature map after morphological processing is closer to the ground truth. The experimental results verify the effectiveness of the morphological processing.

### 3.3. Side Network and Loss Function

To obtain the crack prediction result, the output feature of the morphological network on each scale is inputted to the side network for channel merging and up-sampling. The specific operations are as follows:

(1) The enhanced feature sized of $\frac{W}{4}\times \frac{H}{4}\times 256$ is performed a 1 × 1 convolution operation to obtain the dimension reduction feature with the number of channels being 1. Through two 2 × 2 up-sampling convolution operations, the up-sampled feature with the same size as the original map is obtained. Then, the prediction map of the first scale is obtained by the activation function processing, which is recorded as ${\hat{Y}}_{side}^{\left(1\right)}$.

(2) The prediction result of the second scale is obtained after the enhanced feature sized of $\frac{W}{2}\times \frac{H}{2}\times 128$ is conducted in a 1 × 1 convolution operation, a 2 × 2 up-sampling and activation function processing, which is denoted as ${\hat{Y}}_{side}^{\left(2\right)}$.

(3) The prediction result of the third scale is obtained by only activation function processing, which is expressed as ${\hat{Y}}_{side}^{\left(3\right)}$ with the size of W×H×1.

From the observation of the crack images, it can be seen that the size of the crack takes a small proportion in the image. That is to say, the number of negative samples used for model training is far greater than the number of positive samples. A large number of negative samples will encourage the model to ignore the learning of positive samples, which will lead to a poor prediction of positive samples and a low F1 value. For solving the sample imbalance problem in the model training, the Dice Loss is used to reduce the learning degree of simple negative samples, and thus to improve the segmentation performance. The Dice Loss is calculated as follows.

Given two sets A and B, their Dice similarity coefficient S(A,B) is defined as Equation (7), and its value range is [0, 1]:(7)$$S\left(A,B\right)=\frac{2\left|A\cap B\right|}{\left|A\right|+\left|B\right|}$$
where |A∩B| is the number of elements in the intersection of *A* and *B*, and |A| and |B| represent the number of elements in the set *A* and *B*, respectively. In this paper, *A* and *B* are predicted and true positive sample sets, respectively.

*TP* (True Positive), *TN* (True Negative), *FP* (False Positive) and *FN* (False Negative) are usually used as evaluation indicators in the binary classification problem. Here, A=TP+FP, B=FP+FN and A∩B=TP, so the Dice coefficient S is adjusted as Equation (8):(8)$$S\left(A,B\right)=\frac{2TP}{2TP+FP+FN}$$

Dice Loss $L_{dice}$ is defined as follows:(9)$$L_{dice}=1-S\left(A,B\right)=1-\frac{2TP}{2TP+FP+FN}$$

On the one hand, Dice Loss-based network learning directly uses the segmentation effect evaluation index as the loss function, and on the other hand, a large number of background pixels are ignored in the calculation of the ratio between the intersection and the union. Therefore, the Dice Loss function is chosen to solve the problem of uneven positive and negative samples of crack images and to improve the convergence speed at the same time.

In order to make full use of the multi-scale information of the crack, this paper calculates the loss of the prediction results at each scale and then fuses the loss of the final prediction result to obtain the objective loss function. The design of the multiple loss function is shown in Figure 7.

The final crack prediction result $\hat{Y}$ is obtained by concatenation and 1 × 1 convolution of multi-scale prediction results. The objective loss function $L_{dice}^{sum}$ is the sum of the loss on all scales $L_{dice}^{side}$ and the final prediction loss $L_{dice}^{final}$, which is used to adjust the training of the network model until it converges to the desired value, and thus, the network model is obtained.
(10)$$L_{dice}^{sum}=L_{dice}^{side}+L_{dice}^{final}$$
(11)$$L_{dice}^{side}=\sum_{m=1}^{M}L_{dice}\left({\hat{Y}}_{side}^{\left(m\right)},\;Y_{side}^{}\right)$$
where ${\hat{Y}}_{side}^{\left(m\right)}$ and $Y_{side}^{}$ are the predicted result and the ground-truth of the m−th scale, and M is the total number of scales, which is set as three in the experiment.

Figure 8 shows the crack image, its prediction results on three scales such as ${\hat{Y}}_{side}^{\left(1\right)}$, ${\hat{Y}}_{side}^{\left(2\right)}$ and ${\hat{Y}}_{side}^{\left(3\right)}$ and the final prediction result $\hat{Y}$. $\hat{Y}$ is the final prediction result by performing a 1 × 1 convolution on the concatenation of the features of three scales. As can be seen from Figure 8, shallow features ${\hat{Y}}_{side}^{\left(1\right)}$ and ${\hat{Y}}_{side}^{\left(2\right)}$ pay more attention to the location information of cracks, but there is a relatively large difference between the prediction result and the ground-truth due to the down-sampling operation. ${\hat{Y}}_{side}^{\left(3\right)}$ is more accurate at the details of the crack image due to the combination of deep features and shallow features through cross-layer connections. The final prediction result after fusion is the closest to the ground truth, which proves the effectiveness of multi-scale prediction fusion.

## 4. Analysis of Experimental Results

In order to verify the performance of the proposed crack segmentation network, extensive comparisons are made on five open crack datasets, i.e., CrackTree200 [35], Crack500 [36], GAPs384 [37], AEL [38] and CFD [39]. Further, ablation experiments are used to verify the effectiveness and necessity of each module.

### 4.1. Experimental Dataset and Evaluation Indicators

CrackTree200 [35] is a road crack image dataset that was proposed in 2012, which includes 206 road crack images with resolutions of 800 × 600. The image is rich in shadows and occlusion, and the cracks are slender and diversified in distribution. The numbers of images in the training dataset and the test dataset are 126 and 80, respectively.

Crack500 [36] is a dataset of pavement cracks that was taken with mobile phones in 2016. Each high-definition image was cropped into 16 crack images with a size of 360 × 640. The dataset includes 1896 training images, 348 verification images and 1124 test images.

GAPs384 [37] is the German Asphalt Pavement Distress (GAPs) dataset, which contains various common pavement diseases such as cracks, potholes, patches, etc. The images are characterized by low illumination, oil stains, zebra stripes and other noise, and the image resolution is 540 × 440. The training set and test set contain 409 images and 100 images, respectively.

AEL [38] is a small dataset of crack images collected under various background environments, which consists of 47 training images and 11 test images. 

CFD [39] is a widely used road crack dataset that was photographed by mobile phones, including 118 images with a size of 480 × 320, 95 of which are training images and 23 are test images. Sample labeling is performed at the pixel level.

The development environment used in the experiment included TensorFlow 1.14.0, OpenCV Python 4.5.1.48, cuda10.0.0, cudnn8.04, python 3.6.2, etc. The computing processor was an eight-core Intel Core i7-9700K CPU, and the graphics processor was a GeForce RTX 2080 SUPER.

The objective evaluation indicators used in this paper were the segmentation performance evaluation indicators ODS (Optimal Dataset Scale), OIS (Optimal Image Scale), AIU (Average Intersection over Union) [12] and sODS (simplified versions of ODS) and sOIS (simplified versions of OIS) [18]. OIS is the aggregate F measure of the best threshold in each image in the dataset, and ODS is the best F-measure on the fixed threshold dataset. AIU is the coincidence ratio between the predicted fracture area and the real fracture area. SOIS and sODS are simplified versions of OIS and ODS. The higher the value of the five evaluation indicators, the better the segmentation effect.

### 4.2. Results and Analysis

#### 4.2.1. Comparison and Analysis of Subjective Results

In order to verify the better performance of the proposed algorithm, comparative experiments were conducted with the classical object detection and segmentation algorithms FCN [40], RCF [41], HED [11], FPHBN [12], DAUNet [18] and SPLAC U-Net [23]. Figure 9 shows the original crack image, the ground truth of the crack and the comparison experiment results, where the extracted crack is represented by white pixels, and the background is represented by black pixels.

It can be seen from Figure 9 that the segmentation of cracks of FCN [40] is the most incomplete, especially for the slender cracks appearing in the datasets CrackTree200 [34], AEL [38] and CFD [39]. Compared with FCN [40], the result of RCF [41] is slightly better, but the segmentation results contain a lot of noise, and the boundary of the segmented crack is also fuzzy. The segmented cracks by FPHBN [12] and HED [11] are thicker than the ground truth, and the extracted slender cracks are incomplete, that is, an intact crack is divided into broken parts. DAUNet [18] and SPLAC U-Net [23] can accurately locate cracks; however, the details of extracted cracks are not relatively obvious. As shown in the detection results from columns 3 to 9 of the fifth row in Figure 9, although all the comparison methods detect the interference object as a crack to some extent, the false detection ratio of the proposed method is the minimum. Although the crack segmentation result of the proposed method contains some noise and interference objects, it is the most competitive compared with other methods.

In order to verify the good segmentation performance for the cracks with an abruptly changed shape, Figure 10 shows the three images randomly selected from the CRACK500 [36], CFD [39] and GAPs384 [37] datasets and the segmentation results of the proposed method in this paper. The rectangular boxes in the left image and right image, respectively, represent the selected crack region in the original image and the extracted crack region in the segmentation image. It can be seen from the Figure 10 that the proposed method can easily identify the cracks at the turning, thinning and thickening points.

When the lighting conditions change during the shooting of the crack image, different parts of the same crack may show different colors and brightness values, making the target visually easily misclassified as different objects. Under polarized light, the target discrimination of the depth features extracted by U-Net network would become worse, which inevitably leads to the decline in target segmentation performance. In addition, the bias-light phenomenon also brings some challenges to manual labeling. In order to further verify the good crack segmentation performance of the proposed method in the case of bias-light, three crack images with an obvious light change, a shadow change and under low illumination were selected, respectively, for comparison experiments. Figure 11 shows the original crack images and the segmented cracks by FCN [40], RCF [41], HED [11], FPHBN [12], DAUNet [18], SPLAC U-Net [23] and our method, respectively. As shown in Figure 11, the segmentation results of FCN [40], RCF [41] and HED [11] involved a lot of noise pollution, resulting in poor segmentation performance. Compared to FCN [40], RCF [41], HED [11] and FPHBN [12], the segmentation performance of DAUNet [18] and SPLAC U-Net [23] was better. However, as shown in the sixth and seventh columns of the second row of Figure 11, cracks could hardly be detected by these two methods under low illumination. Although there were a few false alarms and missed detection, the proposed method still achieved the best crack segmentation performance under a variety of conditions. The excellent performance benefitted from the use of white top-hat transformation and black bottom-hat transformation to eliminate the influence of polarized light and enhance the features of the crack.

#### 4.2.2. Comparison and Analysis of Objective Results

In order to verify the good performance of the proposed method on each dataset, Table 2, Table 3, Table 4, Table 5 and Table 6 show the evaluation and comparison results of the ODS, OIS, AIU, sODS and sOIS indicators of some algorithms on the five datasets. From the results in these tables, it can be seen that the FCN [40] had the worst performance on the five datasets, followed by RCF [41]. Both HED [11] and FPHBN [12] had a similar performance, which had a certain performance improvement compared with FCN [40] and RCF [41]. DAUNet [18] and SPLAC U-Net [23] had some improvements compared with the four other networks; however, our method showed absolute advantages in the five datasets. The average ODS, OIS, AIU, sODS and sOIS of the proposed method on the five crack datasets were 75.7%, 73.9%, 36.4%, 52.4% and 52.2%, respectively.

#### 4.2.3. Comparison and Analysis of Ablation Experiments

Ablation experiments were performed to further verify the contribution of each module to the segmentation performance. Figure 12a–f show the original crack image, the ground truth and the segmentation results using only the single-objective loss function, the multi-objective loss function, only the morphological module and a combination of the multi-loss function and the morphological module, respectively. It can be seen from Figure 12 and Table 7 that the segmentation result using only the single-objective loss function was the worst, followed by multi-loss function. The segmentation result of using the morphological module and single-loss function was better, and the segmentation result of the combination of the multi-loss function and the morphological module was the best. Compared with the basic U-Net segmentation network, the segmentation performance indicators ODS, OIS, AIU, sODS and sOIS of our method were increased by 19.8%, 15.2%, 20.9%, 26.6% and 26.0%, respectively.

#### 4.2.4. Comparison and Analysis of Computational Cost

Normally, the FLOPs (floating point operations) and the PARAMS (parameter amount of the neural network) are used to measure the complexity and size of the neural network model, respectively. The FLOPs and PARAMS of our method were about 65.23G and 32.45M, respectively. Compared to the U-Net backbone network, the FLOPs and PARAMS of the combination of the morphological-processing network and the multi-loss function module were only increased by 20.31G and 1.41M, respectively.

We further investigated the execution performance of our crack detection method. Experimental tests were performed on NVIDIA GeForce RTX 2080 SUPER. The crack segmentation ran on average at 3.0 fps, and the computational costs included reading crack images, image preprocessing, feature extraction and fusion and crack segmentation.

## 5. Conclusions

A novel crack segmentation method using a combination of the U-Net backbone network, a morphological-processing network and multi-loss function was proposed in this paper. Aiming at the problem that the existing depth network models lack an effective processing mechanism to eliminate the influence of polarized light, which leads to inaccurate crack segmentation, this paper has designed a morphological-processing network module composed of a white-top hat transformation and a black-bottom hat transformation. This module can correct the influence of uneven lighting and effectively extract the bright cracks on a dark background and the dark cracks on a bright background. In order to avoid the problem of too thin or too thick cracks segmented on a single scale, this paper extracts and fuses the cracks on each scale to obtain more accurate cracks. In addition, a multi-loss function was designed to enhance the robustness of the network model. The experimental results on five crack datasets prove that the proposed algorithm has absolute superiority over several classical crack segmentation algorithms.

## Figures and Tables

**Figure 1 sensors-23-01127-f001:**
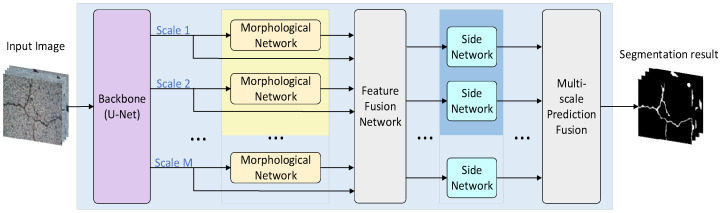
Schematic diagram of crack segmentation method.

**Figure 2 sensors-23-01127-f002:**
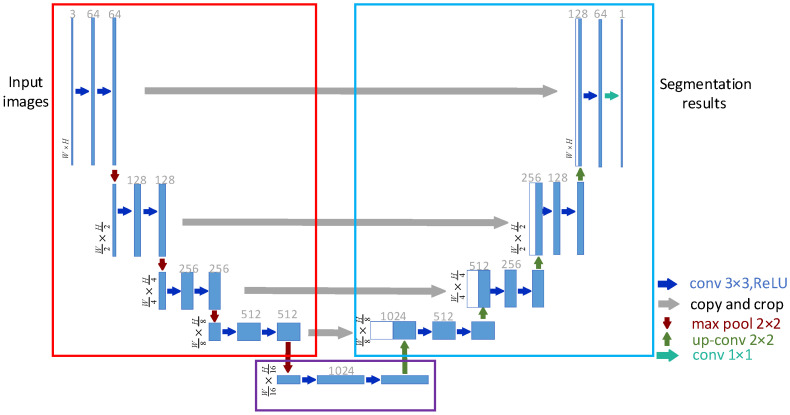
U-Net network structure.

**Figure 3 sensors-23-01127-f003:**
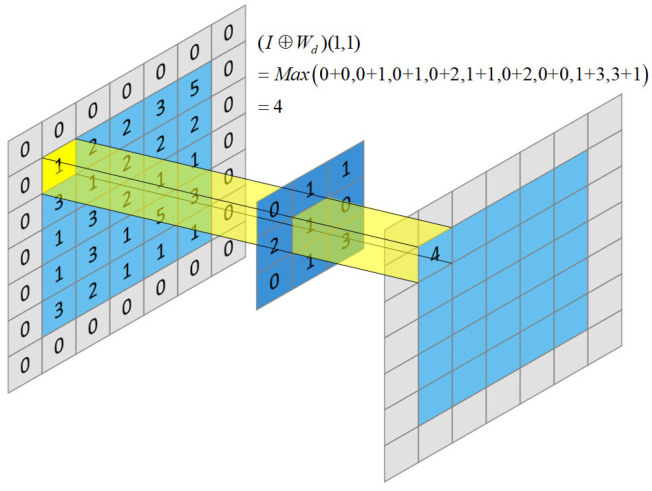
Example diagram of a dilation operation.

**Figure 4 sensors-23-01127-f004:**
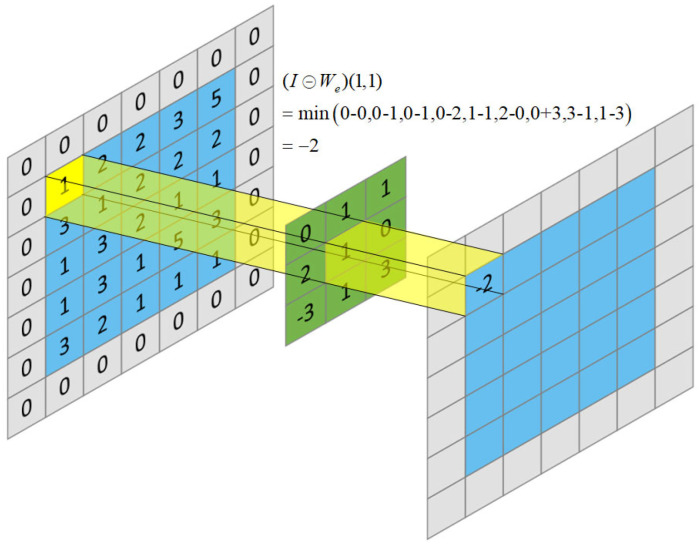
Example diagram of an erosion operation.

**Figure 5 sensors-23-01127-f005:**
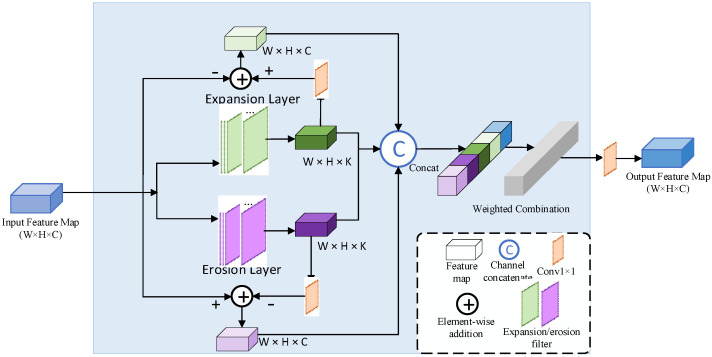
Structural diagram of the morphological network.

**Figure 6 sensors-23-01127-f006:**
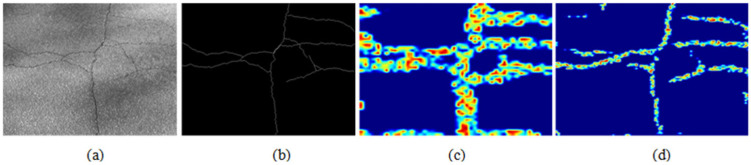
Feature map before and after the morphological network processing: (**a**) original image, (**b**) ground truth, (**c**) feature map before morphological processing, (**d**) feature map after morphological processing.

**Figure 7 sensors-23-01127-f007:**
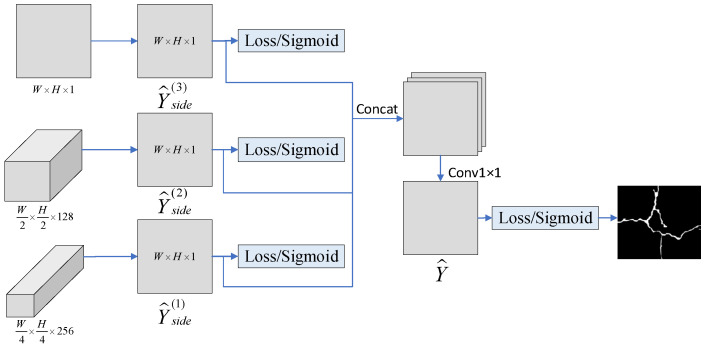
Multi-loss function design.

**Figure 8 sensors-23-01127-f008:**
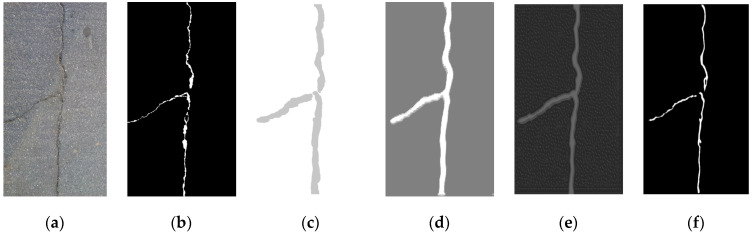
Side network prediction result and final prediction result: (**a**) original image; (**b**) ground truth image; (**c**) ${\hat{Y}}_{side}^{\left(1\right)}$; (**d**) ${\hat{Y}}_{side}^{\left(2\right)}$; (**e**) ${\hat{Y}}_{side}^{\left(3\right)}$; (**f**) $\hat{Y}$.

**Figure 9 sensors-23-01127-f009:**
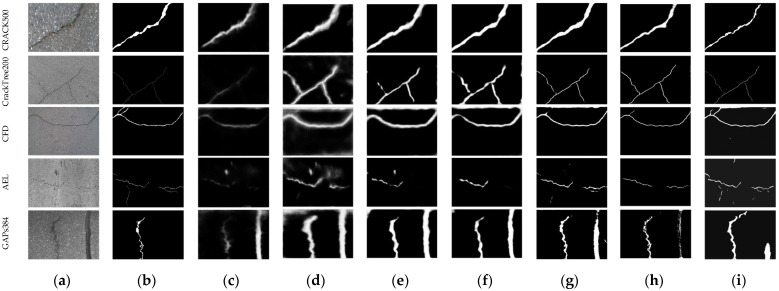
Subjective effect comparison results: (**a**) original images; (**b**) ground truth; (**c**) FCN [40]; (**d**) RCF [41]; (**e**) HED [11]; (**f**) FPHBN [12]; (**g**) DAUNet [18]; (**h**) SPLAC U-Net [23]; (**i**) our method.

**Figure 10 sensors-23-01127-f010:**
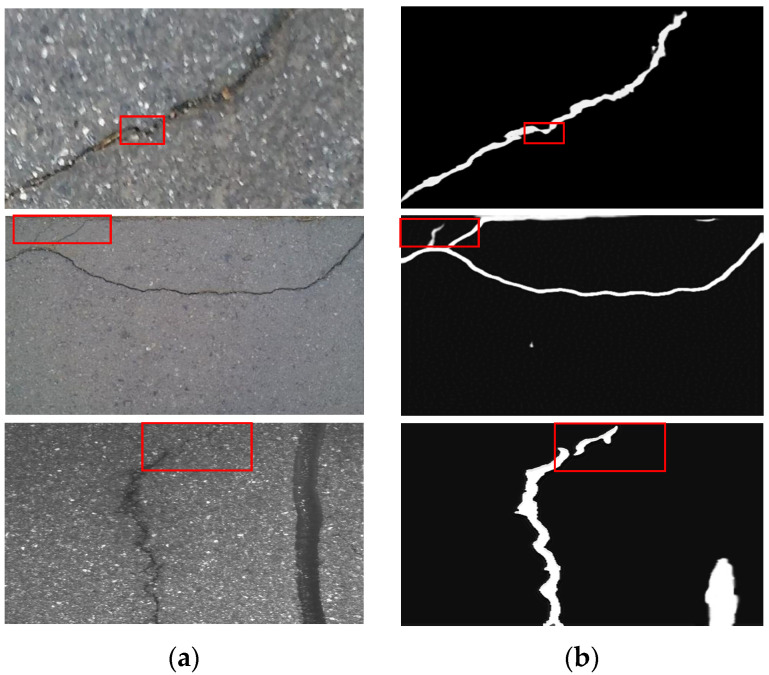
Detail region of crack segmentation results: (**a**) original images; (**b**) segmentation images.

**Figure 11 sensors-23-01127-f011:**
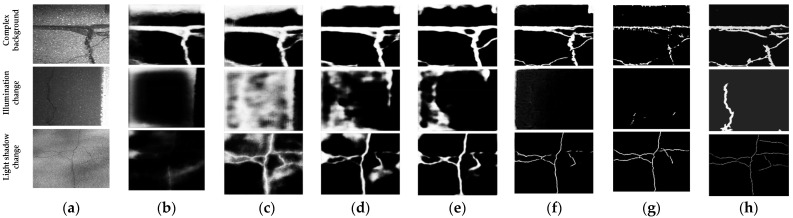
Segmentation comparison results under polarized light: (**a**) original images; (**b**) FCN [40]; (**c**) RCF [41]; (**d**) HED [11]; (**e**) FPHBN [12]; (**f**) DAUNet [18]; (**g**) SPLAC U-Net [23]; (**h**) our method.

**Figure 12 sensors-23-01127-f012:**
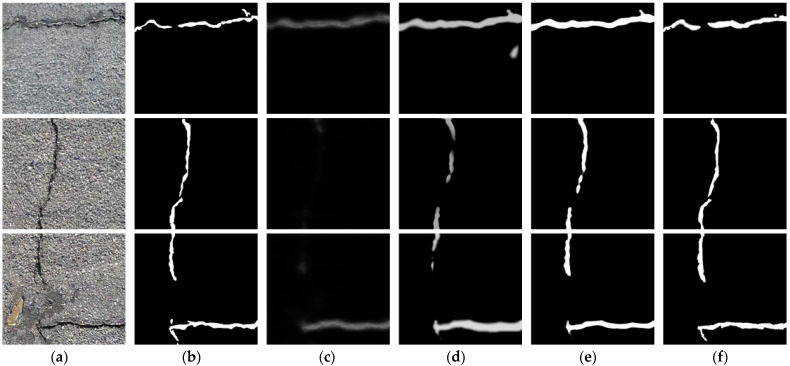
Subjective comparison results of the ablation experiments: (**a**) original image; (**b**) the ground truth; (**c**) the segmentation result using only a single-loss module; (**d**) the segmentation results using only the multi-loss module; (**e**) the segmentation result using only the morphological module; (**f**) the segmentation results with the combination of multi-loss and morphological module.

**Table 1 sensors-23-01127-t001:** The parameter setting of U-Net network structure.

Network Layer Name	Convolutional Layer Parameter Settings	Output Dimension
Conv1-x	$\left[\begin{array}{l} 3\times 3.64\\ 3\times 3.64 \end{array}\right]$	W×H×64
Pooling1	[2×2]	$\frac{W}{2}\times \frac{H}{2}\times 64$
Conv2-x	$\left[\begin{array}{l} 3\times 3.128\\ 3\times 3.128 \end{array}\right]$	$\frac{W}{2}\times \frac{H}{2}\times 128$
Pooling2	[2×2]	$\frac{W}{4}\times \frac{H}{4}\times 128$
Conv3-x	$\left[\begin{array}{l} 3\times 3.256\\ 3\times 3.256 \end{array}\right]$	$\frac{W}{4}\times \frac{H}{4}\times 256$
Pooling3	[2×2]	$\frac{W}{8}\times \frac{H}{8}\times 256$
Conv4-x	$\left[\begin{array}{l} 3\times 3.512\\ 3\times 3.512 \end{array}\right]$	$\frac{W}{8}\times \frac{H}{8}\times 512$
Pooling4	[2×2]	$\frac{W}{16}\times \frac{H}{16}\times 512$
Conv5-x	$\left[\begin{array}{l} 3\times 3.1024\\ 3\times 3.1024 \end{array}\right]$	$\frac{W}{16}\times \frac{H}{16}\times 1024$
Up-Conv1	[2×2.512]	$\frac{W}{8}\times \frac{H}{8}\times 512$
Conv6-x	$\left[\begin{array}{l} 3\times 3.512\\ 3\times 3.512 \end{array}\right]$	$\frac{W}{8}\times \frac{H}{8}\times 512$
Up-Conv2	[2×2.256]	$\frac{W}{4}\times \frac{H}{4}\times 256$
Conv7-x	$\left[\begin{array}{l} 3\times 3.256\\ 3\times 3.256 \end{array}\right]$	$\frac{W}{4}\times \frac{H}{4}\times 256$
Up-Conv3	[2×2.128]	$\frac{W}{2}\times \frac{H}{2}\times 128$
Conv8-x	$\left[\begin{array}{l} 3\times 3.128\\ 3\times 3.128 \end{array}\right]$	$\frac{W}{2}\times \frac{H}{2}\times 128$
Up-Conv4	[2×2.64]	W×H×64
Conv9-x	[1×1.1]	W×H×1

**Table 2 sensors-23-01127-t002:** Segmentation performance comparison results on Cracktree200 dataset.

Method	Year	ODS	OIS	AIU	sODS	sOIS
FCN [40]	2015	0.334	0.333	0.008	N/A	N/A
RCF [41]	2017	0.255	0.487	0.032	N/A	N/A
HED [11]	2015	0.317	0.449	0.040	N/A	N/A
FPHBN [12]	2020	0.517	0.579	0.041	0.095	0.125
DAUNet [18]	2021	0.781	0.805	0.128	0.234	0.276
SPLAC U-Net [23]	2022	0.887	0.894	0.181	0.406	0.427
Our method	2022	0.930	0.932	0.216	0.429	0.430

**Table 3 sensors-23-01127-t003:** Segmentation performance comparison results on Crack500 dataset.

Method	Year	ODS	OIS	AIU	sODS	sOIS
FCN [40]	2015	0.513	0.577	0.379	N/A	N/A
RCF [41]	2017	0.490	0.586	0.403	N/A	N/A
HED [11]	2015	0.575	0.625	0.481	N/A	N/A
FPHBN [12]	2020	0.604	0.635	0.489	0.647	0.591
DAUNet [18]	2021	0.676	0.706	0.565	0.750	0.731
SPLAC U-Net [23]	2022	0.681	0.691	0.583	0.746	0.753
Our method	2022	0.717	0.732	0.592	0.774	0.763

**Table 4 sensors-23-01127-t004:** Segmentation performance comparison results on CFD dataset.

Method	Year	ODS	OIS	AIU	sODS	sOIS
FCN [40]	2015	0.585	0.609	0.021	N/A	N/A
RCF [41]	2017	0.542	0.607	0.105	N/A	N/A
HED [11]	2015	0.593	0.626	0.154	N/A	N/A
FPHBN [12]	2020	0.683	0.705	0.173	0.377	0.372
DAUNet [18]	2021	0.812	0.831	0.370	0.603	0.593
SPLAC U-Net [23]	2022	0.793	0.828	0.383	0.597	0.612
Our method	2022	0.828	0.839	0.403	0.611	0.620

**Table 5 sensors-23-01127-t005:** Segmentation performance comparison results on AEL dataset.

Method	Year	ODS	OIS	AIU	sODS	sOIS
FCN [40]	2015	0.322	0.265	0.022	N/A	N/A
RCF [41]	2017	0.469	0.397	0.069	N/A	N/A
HED [11]	2015	0.429	0.421	0.075	N/A	N/A
FPHBN [12]	2020	0.492	0.507	0.079	0.319	0.283
DAUNet [18]	2021	0.615	0.660	0.223	0.400	0.394
SPLAC U-Net [23]	2022	0.656	0.679	0.235	0.356	0.370
Our method	2022	0.712	0.758	0.205	0.364	0.307

**Table 6 sensors-23-01127-t006:** Segmentation performance comparison results on GAPs384 dataset.

Method	Year	ODS	OIS	AIU	sODS	sOIS
FCN [40]	2015	0.088	0.091	0.015	N/A	N/A
RCF [41]	2017	0.172	0.120	0.043	N/A	N/A
HED [11]	2015	0.209	0.175	0.069	N/A	N/A
FPHBN [12]	2020	0.220	0.231	0.081	0.121	0.156
DAUNet [18]	2021	0.514	0.342	0.217	0.349	0.388
SPLAC U-Net [23]	2022	0.573	0.391	0.389	0.453	0.473
Our method	2022	0.597	0.432	0.403	0.444	0.490

**Table 7 sensors-23-01127-t007:** Ablation experiment results.

U-Net	Morphological Network	Multi-Loss	ODS	OIS	AIU	sODS	sOIS
✓	✗	✗	0.519	0.580	0.383	0.508	0.503
✓	✗	✓	0.598	0.633	0.483	0.559	0.540
✓	✓	✗	0.654	0.659	0.533	0.660	0.604
✓	✓	✓	0.717	0.732	0.592	0.774	0.763

## Data Availability

The datasets CrackTree200, Crack500, GAPs384 and CFD in this paper can be obtained from the following link: https://data.lib.vt.edu/articles/dataset/Concrete_Crack_Conglomerate_Dataset/16625056, accessed on 5 December 2022. The dataset AEL can be obtained at https://tuprd-my.sharepoint.com/:u:/g/personal/tug13683_temple_edu/ESjezwsNLERMpvY85wOEKWkBQKY1A21M1rDhLID11pyRsg, accessed on 5 December 2022.

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
