# Peer review of "A Crack Segmentation Model Combining Morphological Network and Multiple Loss Mechanism"

_sensors, 2023, doi:10.3390/s23031127_

Round 1

Reviewer 1 Report

In this manuscript, a crack segmentation network by combining morphological network and multiple loss mechanism was proposed to solve the problems of high cost, low efficiency and poor detection accuracy in traditional crack detection methods. In addition, compared with state-of-the-art methods, the proposed method achieved better crack segmentation. However, a question should be addressed before publication.

As shown in the fifth row and seventh column of Figure 9, there is obvious missed detection. Therefore, compared with the proposed method, DAUNet is better. I think the conclusion “it is the best compared with other methods” on line 348 is not accurate. Please check it.

Reviewer 2 Report

This paper proposes an image-based segmentation algorithm. My main concern of the paper is for its originality. 

1) Many papers in the field are missing. For example the articles of Prof. Protopapadakis and Doulamis who have receive more than 50 publications

2) In the same field, there exists one work for rust detection in metallic constructions using UNets. There is no mention for such articles

3) there is no comparison with these works 

4) How the images are extracted 

5) Please indicate the impact of the lighting conditions

6) Please reveal the computational cost 

7) what about the use of semi-supervised algorithms.

I am in between reject and major revision, but due to Christmas Holidays I will give one last chance for the authors only if all the previous remarks very carefully addressed. 
